# Accelerated Deep Active Learning with Graph-based Sub-Sampling

**Dan Kushnir**                                          *dan.kushnir@nokia-bell-labs.com*
*Bell Laboratories*
*NOKIA*

**Shiyun Xu**                                          *shiyunxu@sas.upenn.edu*
*Department of Applied Mathematics and Computational Science*
*University of Pennsylvania*

**Reviewed on OpenReview:** *https://openreview.net/pdf?id=iBorskJRrg*

## Abstract

Past years have witnessed the fast and thorough development of active learning, a human-in-the-loop semi-supervised learning that helps reduce the burden of expensive data annotation. Diverse techniques have been proposed to improve the efficiency of label acquisition. However, the existing techniques are mostly intractable at scale on massive unlabeled instances. In particular, the query time of large-scale image-data models is usually linear or even quadratic in the size of the unlabeled pool set and its dimension. The main reason for this intractability is the iterative need to scan the pool set at least once to select the best samples for label annotation.

To alleviate this computational burden, we propose efficient Diffusion Graph Active Learning (DGAL). DGAL is used on a pre-computed Variational-Auto-Encoders (VAE) latent space to restrict the pool set to a much smaller candidate set. The sub-sample is then used in deep architectures, to reduce the query time, via an additional standard active learning baseline criterion. DGAL demonstrates a query time versus accuracy trade-off that is two or more orders of magnitude acceleration over state-of-the-art methods. Moreover, we demonstrate the important exploration-exploitation trade-off in DGAL that allows the restricted set to capture the most impactful samples for active learning at each iteration.

## 1 Introduction

Deep learning has provided unprecedented performance in various semi-supervised learning tasks, ranging from speech recognition to computer vision and natural language processing. Deep Convolutional Neural Networks (CNN), in particular, have demonstrated object recognition that exceeds human performance. However, this success comes with the requirement for massive amounts of labeled data. While data collection at a large scale has become easier, its annotation with labels has become a major bottleneck for execution in many real-life use cases (Settles, 2009; Ren et al., 2021). Active learning provides a plethora of techniques to select a set of data points for labeling which optimally minimizes the error probability under a fixed budget of a labeling effort (see Settles (2009) for review). As such it is a key technology for reducing the data annotation effort in training semi-supervised models.

One of the key caveats in active learning, preventing it from being used on an industrial web scale, is the computational burden of selecting the best samples for annotation at each step of active learning. This complexity is rooted in a variety of important criteria that need to be optimized in active learning. Referred to as the 'two faces of active learning' Dasgupta (2011), the most common selection mechanisms can be categorized into two parts: uncertainty (e.g. Lewis & Gale (1994)) and diversity sampling (e.g Ash et al. (2019a); Sener & Savarese (2017a)). The intuition of the former is to select query points that improve the

model as rapidly as possible. The latter exploits heterogeneity in the feature space, sometimes characterized by natural clusters, to avoid redundancy in sampling. The combination of the uncertainty and diversity criteria has been an important subject of recent works (Yuan et al., 2020; Zhu et al., 2008; Shen et al., 2004; Ducoffe & Precioso, 2018; Margatina et al., 2021; Sinha et al., 2019a; Gissin & Shalev-Shwartz, 2019; Parvaneh et al., 2022; Huijser & van Gemert, 2017; Zhang et al., 2020).

Optimizing for uncertainty or diversity (or both) may require scanning all data at least once, and typically requires methodologies with computational costs that scale quadratic or more in the unlabeled pool size (Bodó et al., 2011; Tong & Koller, 2001; Sener & Savarese, 2017a). A standard active learning cycle repeats the time-consuming model (re-)training and query selection process multiple times. In many cases, these cumbersome repetitions render active learning impractical on real large-scale data sets. Even a single feed-forward process in a deep learning network for uncertainty calculation (e.g. Gal et al. (2017)) may impose a significant delay in query time. It may scale non-linearly in the dimension of the data (e.g. number of pixels) for each query candidate in many deep network architectures, where fully connected layers exist. To this end, only a few approaches have been suggested to overcome this bottleneck in the context of deep learning. Most notable is SEALS Coleman et al. (2022a), which improves the computational efficiency of active learning and rare class search methods by restricting the candidate pool for labeling to the nearest neighbors of the currently labeled set instead of scanning over all of the unlabeled data. The restricted set is given as an input to the task classifier for a second selection step, and using a second basic uncertainty criterion, a final query set is selected for annotation.

We identified that the SEALS criterion of selecting $k$-nearest neighbors to the restricted pool does not address the diversification criterion in query selection. It also does not capture the exploration-refinement transition which improves active learning tremendously. Therefore, we propose a different algorithm for the selection of the restricted pool set based on a graph diffusion algorithm inspired by (Kushnir, 2014; Kushnir & Venturi, 2023). We refer to it as Diffusion Graph Active Learning (DGAL). In DGAL, the proximity graph is computed **only once** for a latent Variational Auto Encoder (VAE) Kingma & Welling (2013) representation space which is trained (without supervision) once prior to the annotation cycles. This graph representation is then used for a label diffusion process to select the most diversified and uncertain candidates in time that is log-linear in the data size (Appendix 4.4). The graph, computed only once, allows faster linear query time in each diffusion process, unlike SEALS Coleman et al. (2022a), whose query time keeps growing by a factor dependent linearly on the number of neighbors $k$.

As seen in the overview of our method in figure 1, our method is comprised of two components: a representation component in which a VAE training occurs once and is used to generate a proximity graph that is then shared in the iterative stage of the active learning component. We note that other representation spaces may also be considered for graph construction. A label diffusion process on the graph is used throughout the active learning iterations, in the second component, to select a restricted and small set of candidates which is fed into the classification neural net for the final query selection (e.g. uncertainty sampling (Lewis & Gale, 1994), margin (Scheffer et al., 2001), etc.). As shown in our paper the restriction to a smaller set via our graph construction accelerates active learning while achieving state-of-the-art or better accuracy.

The graph-based diffusion algorithm used in DGAL enhances an important criterion in active learning referred to as the exploration-exploitation (or exploration-refinement criterion). Exploration addresses a stage in active learning in which data are sampled and annotated to first map decision boundaries in the data. On the other hand, exploitation takes the so far detected boundaries and samples points around them to further localize the boundaries. At the early stages of AL, an exploratory strategy typically yields better gains in accuracy over boundary refinement. However, once all boundaries are detected, typically a refinement stage provides better accuracy gains over further exploration. This important trade-off is not leveraged in standard simple AL criteria such as probabilistic uncertainty sampling or diversification per se. In particular, a k-nearest-neighbors to the training set (Coleman et al. (2022a)) also does not reflect this trade-off. We demonstrate in this paper that our combined graph methodology for pool set restriction improves AL query time and scales it to large-scale data sets. We summarize our contributions below:

- We propose DGAL, a two-step active learning algorithm, that starts with restricting the pool set to a smaller set using a graph-diffusion process and then uses the restricted set to perform deep active learning efficiently.

- DGAL achieved orders of magnitude acceleration in query time compared to state-of-the-art (SOTA) active learning schemes, while maintaining the most competitive accuracy.

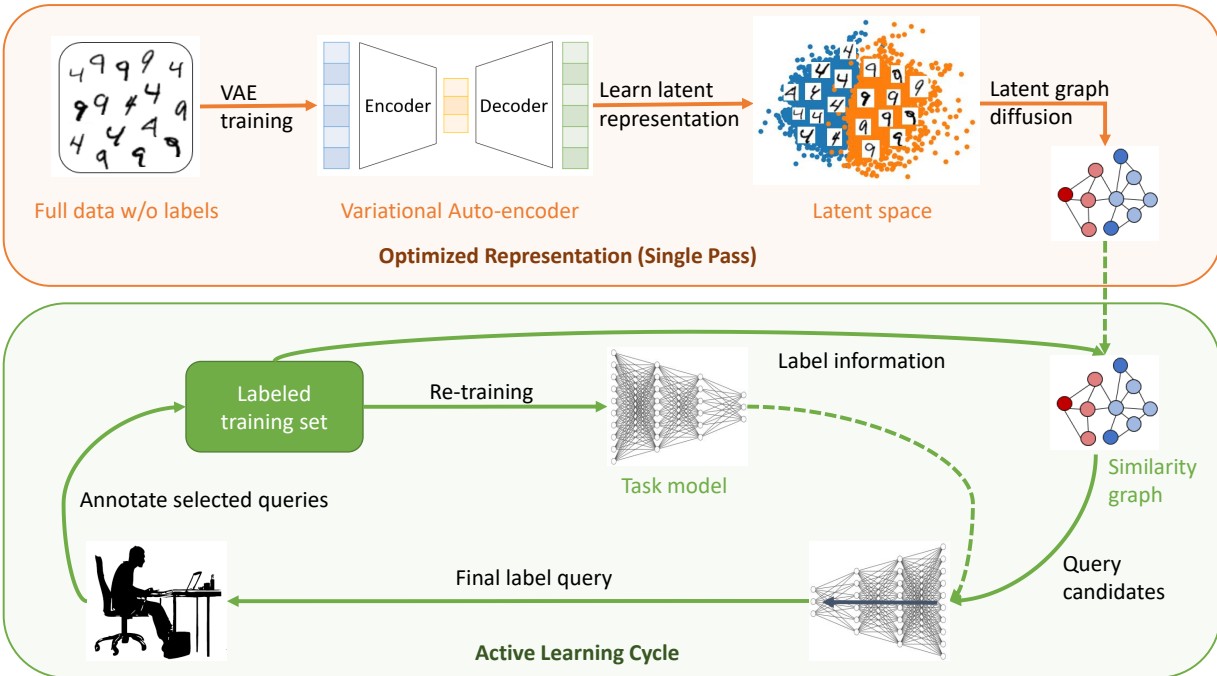

Figure 1: **Overview of the DGAL approach**. A Representation component that involves a single pass on the data for deriving a VAE-based latent graph representation. An Active Learning iterative component that uses the VAE latent representation graph in conjunction with existing label information to select a restricted set of query candidates. The set of candidates is fed into a task neural net, where a second criterion is used to select the final label queries for annotation.

## 2 Related research

**Efficient Active Learning.** With the increasing availability of large-scale unlabeled datasets, traditional active learning methods are too computationaly demanding to apply. This problem has motivated the development of more compact, efficient AL algorithms to cope with large-scale data sets/models. In SEALS Coleman et al. (2022a) the candidate pool is restricted to the k-Nearest Neighbours (KNN) of so-far-labeled instances, to avoid the computational burden of batch selection with large pool sets. However, the KNN set is typically very similar to the already labeled training data and therefore its information content for improving the classifier is low. This lack of diversification and exploration in SEALS yields sub-optimal accuracy. Moreover, the number of nearest neighbors being fed later to the tasks neural net is growing by a factor of $k$ on each query step. This leads to an increasingly higher query time when the restricted pool is fed into the task network. Xie et al. (2021) propose knowledge clusters extracted from intermediate features over pre-trained models by a single pass. Ertekin et al. (2007); Doucette & Heywood (2008) proposed to actively subsample unlabeled candidates under imbalanced datasets in general learning schemes, different than deep learning. Generating informative samples Mayer & Timofte (2020); Huijser & van Gemert (2017); Zhu & Bento (2017) saves time from the unlabeled data acquisition aspect. Small but well-trained auxiliary models have been used to select data to reduce the computational cost of extracting feature representation from the unlabeled pool (Yoo & Kweon, 2019; Coleman et al., 2019).

**Graph-based Semi-Supervised Learning (GSSL)** Semi-supervised learning (SSL) Zhu (2005) exploits the information of both the labeled and unlabeled datasets to learn a good classifier. As a form of SSL, active learning automates the process of data acquisition from a large pool of unlabeled datasets for annotation to achieve the maximal performance of the classifier (usually) under a limited budget. GSSL (Zhu, 2005; Song et al., 2022; Zha et al., 2009) is a classic branch of the SSL that aims to represent the data in a graph such that the label information of the unlabeled set can be inferred using the labeled data. A classic and solid technique is label propagation or Laplace learning Zhu et al. (2003), which diffuses label information from labeled sets to unlabeled instances. Notably, the computational complexity of a typical label propagation algorithm is only linear in the size of the data, which renders them efficient choices for learning.

The success of label propagation hinges on an informative graph that retains the similarity of the data points. Due to the volatile property of image pixels, i.e., unstable to noise, rotation, etc., feature transformations Lowe (1999); Bruna & Mallat (2013); Simonyan & Zisserman (2014) are usually applied to build good quality graphs. Past research Doersch (2016); Kingma et al. (2019); Mei et al. (2019); Miller et al. (2022); Calder et al. (2020) has shown that the VAE can generate high-quality latent representation of data for feature extraction and similarity graph construction. We utilize these properties in our DGAL framework.

**Generative Models in Active Learning.** Deep generative models have been used to learn the latent representation of data in both semi-supervised and unsupervised learning (Kingma et al., 2019; 2014). Except for constructing similarity graphs in GSSL, they can also be exploited to generate adversarial data/models for more efficient and robust AL. For example, Sinha et al. (2019a) proposed a task-agnostic model that trains an adversarial network to select unlabeled instances that are distinct from the labeled set in the latent space of a Variational Auto-Encoder (VAE). The DAL Gissin & Shalev-Shwartz (2019) selects samples in a way such that the labeled and unlabeled sets are hard to distinguish in a learned representation of the data. Miller et al. (2022) embeds the Synthetic Aperture Radar (SAR) data in a latent space of VAE and applies a GSSL method on a constructed similarity graph in this feature space. Pourkamali-Anaraki & Wakin (2019) aims at finding a diverse coreset in a representative latent space using K-Means clustering.

## 3 Problem setup and preliminaries

### 3.1 Active Learning

We consider a data set $D \in \mathbb{R}^d$ of cardinally $[n]$. $D$ can split into a labelled set $D_l$ which represents a labeled subset of $D$, and an unlabeled subset $D_u$. $C$ denotes the number of classes. Fixing a batch size $B$, we seek for the most 'informative' subset $D^\star \in D_u$ to be annotated from an unlabeled pool set given a limited budget for annotation.

**Active Learning Problem Statement**. Let $f_\theta$ be a classifier: $f_\theta : \mathbb{R}^d \to \Delta_C$, ($\Delta_C$ denotes the space of probability measures over $[C] = \{1, ..., C\}$). Assume

---

**Algorithm 1** A general active learning algorithm

**Input:** Labeled data $D_l$, unlabeled pool $D_u$,
batch size $B$, maximum round $R$, task model $f_\theta$
Initialize $D_l$ by random sampling and train $f_\theta$ on
**for** $i = 1$ **to** $R$ **do**
   query $D_Q \leftarrow \boldsymbol{QueryStrategy}(f_\theta, D_u, B)$
   update $D_l \leftarrow D_l \bigcup D_Q$, $D_u \leftarrow D_u \setminus D_Q$
   train model $f_\theta$ on new labeled pool $D_l$
**end for**

---

data are sampled *i.i.d.* over a space $\mathcal{D} \times [C]$, denoted $\{x_i, y_i\}_{i \in [n]}$. We would like to find a subset $D_Q^\star \subset D_u$ of cardinality $B$ such that for a classifier $f_\theta$ trained with $D_l \bigcup D_Q^*$

$$D_Q^\star = \underset{D_Q : |D_Q| = B}{\arg\min} \; \mathbb{E}_{x, y \in D_u}[\mathbb{I}\{f_\theta(x) \neq y\}] \tag{1}$$

gives the minimum expected error. Clearly, finding $D_Q^* \subset D_u$ that minimizes (1) is not possible without knowing the labels. Hence various active learning strategies to approximate $D_Q^*$ have been developed in the literature. To summarize we provide the general active learning framework in algorithm 1.

**Query Time**. Query time is the time measured between the activation of the query strategy and its return with a selected set, i.e. first line in the loop of Algorithm 1.

### 3.2 VAE-based data representation

Latent variable models such as the VAEs are well-acknowledged for learning representations of data, especially images. VAE is a generative neural network consisting of a probabilistic encoder $q(z|x)$ and a generative model $p(x|z)$. The generator models a distribution over the input data $x$, conditioned on a latent variable $z$ with prior distribution $p_\theta(z)$, for simplicity we omit the subscript $\theta$. The encoder approximates the posterior distribution $p(z|x)$ of the latent variables $z$ given input data $x$ and is trained along with the generative model by maximizing the evidence lower bound (ELBO):

$$ELBO(x) = E_{q(z|x)}[\log(p(x|z)] - KL(q(z|x)||p(z))$$

where KL is the Kullback–Libeler divergence and $\log(p(x)) \geq ELBO(x)$.

To get a representative latent space from data (without label information), we use a ResNet18 (He et al., 2016) based encoder for large datasets and a CNN-based VAE for all other datasets. In the first term, $E_{q(z|x)}[\log(p(x|z)] = \frac{1}{n}\sum_i^n (x_i - x_i')^2$ where $x_i'$ is the $i$-th (out of $n$) reconstructed image. We use the MSE loss for the first term, i.e. the reconstruction loss. Next, we construct a proximity graph from the latent representation to be used within a graph diffusion process and label acquisition.

Figure 2 visualizes a 2D projection of a CNN-VAE 5D representation of digits '4' and '9' from the MNIST dataset. To demonstrate the important semantics of this representation for active learning, we focus on the decision boundary between digits. we observe samples of '4's that are similar to samples from the '9' class along the boundary.

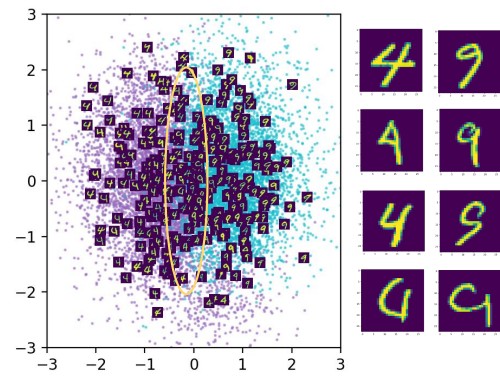

Figure 2: A VAE representation of digits '4' and '9' from MNIST.

## 4 Methodology

Below we explain the various components of VAE-DGAL and then connect them to provide the overall DGAL scheme. Active learning is used in DGAL in a similar scheme to what is proposed in algorithm 1. However, we are using an active criterion in two components: first, we use a very efficient graph-diffusion-based selection criterion Kushnir & Venturi (2023) in the VAE latent space to restrict the pool set to a much smaller set of candidates. In the second component, we use a second acquisition criterion to select from the set of query candidates the final query set for annotation. The second selection criterion may be computationally intensive due to its nature, but also because it requires a feed-forward transmission. However, when applied to a smaller, restricted set can be executed extremely fast.

We provide the pseudo-code of our method in Algorithm 2. We note that the input to VAE-DGAL includes the labeled and unlabeled pool set, the VAE architecture, and the task classification network $f_\theta$. After training the representation model $g$, we initiate active learning cycles which include i) diffusion-based selection for restricting the pool set, ii) feeding the restricted set to the task network $f_\theta$ and, iii) using a basic active learning criterion to select the final set. After annotating the final set, $f_\theta$ is retrained. The time complexity analysis showing a log-linear complexity can be found in Section 4.4.

The VAE representation space can be replaced by other representations. Its advantage is in being derived via an unsupervised method which requires no labels and can be performed only once, prior to label acquisition. The underlying assumption is that the representation space bears a structure that correlates with the class function, and therefore can be used to restrict the pool set to a smaller, yet, impactful set of candidates for annotation.

### 4.1 Diffusion on graphs

Consider the optimized latent representation $g(D) = Z$, where $g : \mathbb{R}^d \rightarrow \mathbb{R}^k$, and $k$ is the dimensionality of the latent space $Z$. $Z$ may also be divided into a mapping of the labeled and unlabelled set as $Z_l$ and $Z_u$,

respectively. In this latent space, we construct a weighted KNN proximity graph $G = (V, E)$, where the nodes $V$ correspond to the latent space representation of the data points $z_i \in Z$, and the edges $E$ correspond to the pairs $(v_i, v_j)$ corresponding to $(z_i, z_j)$ who are neighboring in $Z$. The edge weights are computed as

$$W_{ij} = m\left(-\frac{\rho(z_i, z_j)}{\sigma_{ij}}\right) \mathbb{I}\{j \in N(i)\}$$

with $m : (z_i, z_j) \to \mathbb{R}_+$ as a similarity metric, $\rho$ as a distance metric, $\sigma_{ij}$ as a local scaling factor, and $N(i)$ as the $K$-NN neighbourhood of node $i$ in the latent space. We define a graph transition matrix $M$ by

$$M \doteq \Lambda^{-1} W, \tag{2}$$

where $\Lambda = diag(\sum_j W_{ij})$. $M$ stands for the transition probabilities of a Markov random walk on $G$.

We construct a label diffusion framework as a Markov process to propagate the label information from the labeled set $D_l$ to the unlabelled set $D_u$. The transition probability of a step from state $i$ to state $j$ is $M_{ij}$. W.l.g. consider a binary classification and the labeling function $y(z) \to \{-1, 1\}$ to facilitate the following: we associate the classification probability $p(y(z_i) = 1|z_i)$ with a $t$-step hit probability - $p_t(y(z) = 1|i)$ of a random walk from $z_i$ to a training sample $z$ with label 1. We can therefore predict the label '1' to $z_i$ in $Z_u$ based on the random walk probability $p_t(y(z) = 1|i)$. $p_t(y(z) = 1|i)$ can be derived by the recursive relation

$$p_t(y(z) = 1|i) = \sum_j p_{t-1}(y(z) = 1|j)p_{ij}. \tag{3}$$

Let $\chi_i = 2p_t(y(z) = 1|i) - 1 \in [-1, 1]$ denote an approximation to the binary label function of a node $v_i$. We also denote $\chi_u$ and $\chi_l$ as the entries corresponding to the unlabeled and labeled node sets in $\chi$, respectively. In matrix form we can rewrite (3) for $\chi$

$$\chi_u = [\Lambda_{uu}^{-1}W_{ul}|\Lambda_{uu}^{-1}W_{uu}]\begin{pmatrix}\chi_l \\ \chi_u\end{pmatrix}, \text{ where } \Lambda = \begin{pmatrix}\Lambda_{ll} & 0 \\ 0 & \Lambda_{uu}\end{pmatrix}, W = \begin{pmatrix}W_{ll} & W_{lu} \\ W_{ul} & W_{uu}\end{pmatrix}. \tag{4}$$

We can rewrite (4) the graph Laplacian $L = D - W$ in the system

$$L_{uu}\chi_u = W_{ul}\chi_l \iff L_{uu}\chi_{uu} = -L_{ul}y_l \tag{5}$$

Equation (5) above can be solved via the iteration (Chapelle et al., 2009):

$$\chi_i^{(t+1)} = \frac{1}{L_{uu,ij}}\left(-(L_{ul}y_l)_i - \sum_{j \neq i} L_{uu,ij}\chi_j^{(t)}\right), \tag{6}$$

where the superscript corresponds to the iteration index. Equation (6) is transducing, at time step $(t+1)$, a label $\chi_i^{t+1}$ to $\chi_i$ as a weighted average of the labels of its neighbors at time step $t$.

Now consider the multi-class case, $\chi$ is now an $n \times C$ matrix. For the signal column vector corresponding to the class $c \in C$, the diffusion process initializes with

$$\chi_{ic}^{(0)} = \begin{cases} 1 & \text{if } z_i \in Z_l \text{ and } c = y_i \\ -1 & \text{if } z_i \in Z_l \text{ and } c \neq y_i \\ 0 & \text{if } z_i \in Z_u. \end{cases} \tag{7}$$

The labels are propagated to $\chi_u$ gradually for $t$ steps. At the $(t+1)$-th step,

$$\chi_{ic}^{(t+1)} = \begin{cases} 1 & \text{if } z_i \in Z_l \text{ and } c = y_i \\ -1 & \text{if } z_i \in Z_l \text{ and } c \neq y_i \\ (M\chi_{:,c}^t)_i & \text{if } z_i \in Z_u. \end{cases} \tag{8}$$

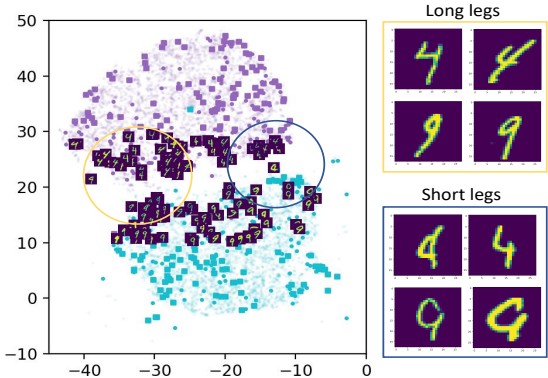

Figure 3: The DGMG selection of digits 4s and 9s (from MNIST) in an embedding space of ResNet18 well-trained with full data and labels.

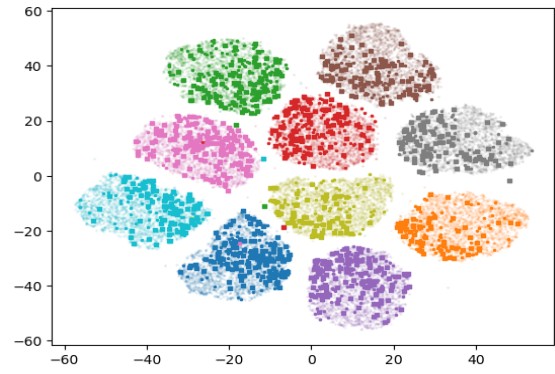

Figure 4: The DGMG selection of CIFAR10 in a ResNet18 embedding space. The dark dots are selected points by DGMG.

## 4.2 Query criterion

To this end, the diffusion process can be used to probe the most uncertain points. Most uncertain points maximize the gradient of the loss function in expectation and therefore will reduce the expected error in the model in Equation (1). Active learning, therefore, aims to query points of highest uncertainty.

The matrix $\chi^{(T)}$ of propagated values at time $T$ can be interpreted as **uncertainties** measured by the absolute value $|\chi_{c,i}^{(T)}|$. Specifically, the absolute value magnitude represents a measure of uncertainty on whether vertex $i$ belongs to class $c$. The magnitude can be used to select the new batch to query as

$$D_Q^{sub} = \{x_i : \arg\min_{z_i \in Z_u}^B \min_{c \in [C]} \|\chi_{c,i}^{(T)}\|\}, \tag{9}$$

where $\min^B$ denotes the $B$ smallest elements.

We demonstrate the exploration-refinement trade-off of (9) in Figure 3, with a subset of MNIST (LeCun et al., 1998) representation in a trained ResNet18 (He et al., 2016). A higher query rate is observed close to the decision boundary after the overall clusters have been explored (queried samples are in bold round points). We also demonstrate that the query selection along the boundary captures '4's and '9's that have similar shapes, 'leg' sizes, and orientations. These samples whose similarity can be best learned via annotation are automatically selected by our criterion (9) for annotation.

In Figure 4 we plot the latent representation of CIFAR10 (Krizhevsky et al., 2009). We mark the points selected by the first five rounds as dark, round dots and data selected by the next five rounds as rectangular, dark dots. Different color stands for different classes and points with lighter colors are the latent representation of all the CIFAR10 data. Here a similar trend is captured showing the sample selection starts with exploration and then tends to the refinement of decision boundaries between clusters of different classes.

**Exploration and Refinement.** Our query criterion coupled with the diffusion process allows us to explore the data set at the early stages of active learning and switch to refinement when exploration has saturated. To understand this mechanism we show in the following that the diffusion iterant $\chi^t$ converges to the second eigenvector $\phi_2$ of the graph's Laplacian as $t \to \infty$. Asymptotically, $\phi_2$ provides a relaxed solution to the minimal normalized cut problem, where the cut corresponds to the decision boundary between the two classes in $G(V, E, W)$.

At early stages of label acquisition low magnitude entries in $\chi$ correspond to data points that are unreachable from the training set via diffusion and need to be explored. At later stages, all unlabelled data points $X_u$ are reachable via diffusion from the labeled set $X_l$. At this stage low magnitude entries correspond to the transition between the two classes -1 and 1. These nodes capture the eigenvector's transition from negative to positive entries. Therefore, sampling these points for annotation corresponds to the refinement of the

decision boundary. We provide the main theoretical result on the convergence of $\chi^{(t)}$ and refer for further details in Kushnir & Venturi (2023).

**Lemma 4.1.** *Let $\lambda_1, ..., \lambda_n$, $\phi_1, ..., \phi_n$ be the solutions to the system: $L\phi = (1-\lambda)D\phi$. Then $\chi^{(t)}$ converges to $\phi_2$ via the iteration (6) with $M$, as $t \to \infty$.*

### 4.3 Deep active learning with the task classifier

At the last stage of the algorithm we input the restricted subset $D_Q^{sub}$ into the task net and using a baseline active learning criterion we select the final set $D_Q$ to be sent for annotation. Feeding pool data into the network poses a significant computational cost. However, since the restricted set $D_Q^{sub}$ is significantly smaller than $D_u$ we observe a significant speedup in the query time. In our experiments, we used two simple baselines to demonstrate the speed of DGAL. However, any other deep active learning criterion can be used, including another diffusion-based criterion (Kushnir & Venturi, 2023).

**Least Confidence.** Confidence sampling (Lewis, 1995): $x_q = \arg\max_x(1 - f_{\hat{c}}(x))$, where $f(x)$ is the prediction probability extracted from the output layer and $\hat{c} = \arg\max f(x)$. Denoted as **DGLC**.
**Margin.** Margin sampling (Scheffer et al., 2001): $x^\star = \arg\min_x(f_{\hat{c}_1}(x) - f_{\hat{c}_2}(x))$, where $\hat{c}_1, \hat{c}_2$ are the first and second most probable prediction labels respectively. Denoted as **DGMG**.

---

**Algorithm 2** The DGAL strategy

**Input:** Labeled data $D_l$, unlabeled pool $D_u$, the initial VAE model $g$, the task model $f_\theta$, $T$, $K$, batch sizes $B_{1,2}$, round $R$, a query strategy $Q$
**Train** $g$ with the full dataset (without label information)
**Build** graph $G = (V, E, W)$ from the latent space $g(D) = Z$
**for** $r = 1$ **to** $R$ **do**
    Initialize $\chi^{(0)}$ based on $D_l$
    **for** $t = 1$ **to** $T$ **do**
        $\chi^{(t)} \leftarrow M\chi^{(t-1)}$
        assign $\chi^{(t)}$ with training labels
    **end for**
    **Query a restricted set of size $B_1$ :**
    $\mathbf{D}_Q^{sub} = \{x_q : z_q = \arg\min_{i \in Z_u, c \in [C]} |\chi_{i,c}^{(T)}|\}$
    **Active sub-sampling of batch $D_Q^{sub}$ with criterion $Q$**
    **and network $f_\theta$:** $D_Q \leftarrow Q(f_\theta, D_Q^{sub}, B_2)$
    **Annotate** $D_Q$
    **Update** $D_l \leftarrow D_l \bigcup D_Q$, $D_u \leftarrow D_u \setminus D_Q$
    **Train** $f_\theta$ with $D_l$
**end for**

---

### 4.4 Computational complexity analysis and parameter selection

The running time analysis is composed of three parts. The first part includes the computation of the $K$-NN graph. The $K$-NN proximity search computational cost can be reduced from the naive search cost by using procedures for $K$-NN search based on KD (Bentley, 1975) or ball trees (Omohundro, 1989). Such methods have complexity $O(dN \log N)$, with $d$ as the dimension of the space. Other alternatives include approximate search (Datar et al., 2004). In the second part, the diffusion vector $\chi$ is multiplied by the transition matrix. Addressing its sparsity as $O(KN)$ non-zero entries, this operation scales linearly in $N$ as $O(TKN)$. $T$ and $K$ determine the level of confidence imposed by the diffused training set over the unlabeled set. Higher $K$ imposes strong confidence in the current labeling hypothesis but renders the diffusion more exhaustive. Similarly, a large number of iterations $T$ may result in an overly smoothed (and less informative) signal $\chi^{(T)}$. During exploration, large $T$ imposes a hypothesis that may be locally correct but is far from being globally reliable. In our experiments, we use $T \simeq \log_K N$ to cover most of the graph via diffusion: assuming a diameter-balanced graph (S. Miklavic, 2018). The cover requires $O(\log_K N)$ iterations if the labeled set is small (i.e. $|X_l| \approx O(1)$, as typical in active learning settings). We use $K$ sufficiently high to allow graph connectivity. This parameter selection leads to a diffusion process that scales as $O(KN \log_K N)$.

Finally, a batch of the smallest soft-labels needs to be queried. This requires a quick-sort to be applied to the soft labels magnitude, which scales $O(N \log N)$. We conclude that the running time of DDAL is $O(dN \log N + KN \log_K N + N \log N)$, whereas for a constant $K$ can be further simplified to $O(dN \log N)$. We note that the number of units in the penultimate layer $d$ is typically small.

**Additional parameters:** The batch size $B$ is set at no more than 500 (see batch size for diffusion algorithm in Kushnir & Venturi (2023). The number of epochs is set with a stopping criterion for the convergence of the loss function.

## 5   Experiments

Our experiments validate our goal to reduce the query time while maintaining the highest accuracy. We report DGAL's improved query time vs. accuracy trade-off and compare it with pivotal SOTA baselines. Additionally, we provide classical active learning empirical analysis of the trade-off between the number of queried data points and accuracy, and we provide average query times, for all baselines and data sets. We provide an ablation study with VAE-SEALS (see Algorithm 3) and with random versions of pool set restriction. Training details are provided in table 1 in the appendix.

### 5.1   Benchmarks

**Random**: selects samples uniformly at random for annotation. **Confidence based methods**: A set of conventional selection strategies includes Least-Confidence (LC) (Lewis, 1995), Margin (Scheffer et al., 2001; Luo et al., 2005) and Entropy (Holub et al., 2008). In LC, the instance whose prediction is least confident is selected; Margin selects data that has minimum difference between the prediction probability of the first and second most confident class labels. Entropy selects samples that are the most uncertain overall class probabilities on average. **BADGE** (Ash et al., 2019a,b): Selects points based on their gradient magnitude and diversity. **CoreSet** (Sener & Savarese, 2017a,b): An algorithm that selects a core-set of points that $\lambda$-cover the pool set. **SEALS** (Coleman et al., 2022a,b): restricts the candidate pool to the nearest neighbors of the currently labeled set to improve computational efficiency. We examine two versions of SEALS, one that updates the feature extractor, and another that sets it fixed using a VAE. **GANVAE** (Sinha et al., 2019a,b): A task-agnostic method that selects points that are not well represented in the pool, using a VAE and an adversarial network.

**Data sets and setting.** Experiments are conducted on multiple data sets to evaluate how DGAL performs in comparison to SOTA benchmarks. We also perform an ablation study. We experimented with benchmark data sets MNIST LeCun et al. (1998), EMNIST Cohen et al. (2017), SVHN Netzer et al. (2011), CIFAR10 Krizhevsky et al. (2009), CIFAR100 Krizhevsky et al. (2009), and Mini-ImageNet Ravi & Larochelle (2017) data sets. We include classic networks CNN LeCun et al. (1998), ResNet18 He et al. (2016), ResNet50 He et al. (2016), and ViT-Small Dosovitskiy et al. (2020).

### 5.2   DGMG vs. Benchmarks

**Query time vs. test accuracy.** In Figure 5 we report accuracy per total query time for a fixed number of queries. We observe for all data sets that DGMG's accuracy per query time is by far higher than SOTA. Moreover, we emphasize its ability to reach the highest accuracy vs. the random baseline, which is the fastest method with essentially close to zero query time (approx. $10^{-3}$ secs.). It can be seen that the acceleration of DGAL is of an order of x1000 with respect to the worst benchmark method and x100 concerning the next best-performing method. These results emphasize the advantage of the DGALs graph diffusion approach in selecting more impactful samples for the pool set restriction, in particular, the advantage over using the k-nearest-neighbors to the existing training set, as proposed in SEALS. The KNN criterion does not select a diversified set or even focus on decision boundary refinement in the restricted set. Consequently, its accuracy gain in each query step is lower. Additionally, the ever-growing candidate size in SEALS (by a factor of the number of neighbors) increases the total query time.

We note that in our experiments for Figure 5, the SEALS feature extractor is trained with labeled data before any acquisition cycles by using a part of the labeled data to implement the KNN data structure. The remaining data serves for pool acquisition. Such an approach does not apply to active learning where initial labels may not even exist unless transferred from a pre-trained model, as suggested in Coleman et al. (2019). To provide a consistent benchmark, i.e. applying SEALS in the same way for all datasets, we adjust the algorithm by feeding the datasets into the trained task model to get an embedding representation as the

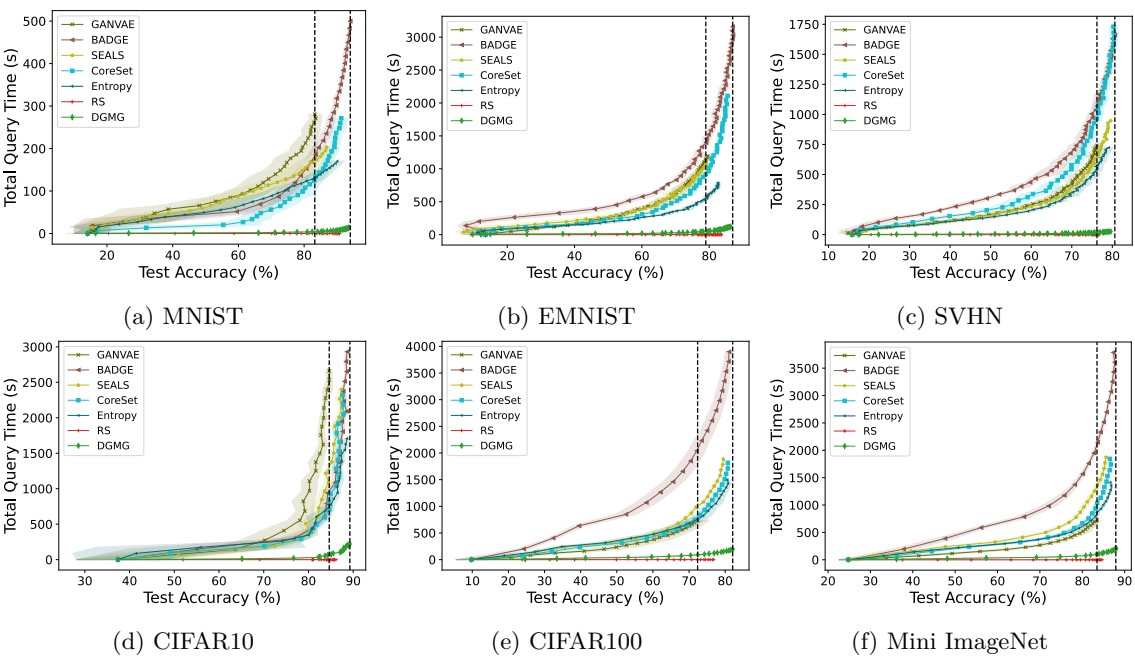

(a) MNIST  (b) EMNIST  (c) SVHN

(d) CIFAR10  (e) CIFAR100  (f) Mini ImageNet

Figure 5: Plots of total query time vs test accuracy for 6 datasets over 6 benchmarks. The vertical lines capture the highest and lowest test accuracy among all methods.

feature extractor, which adds up query time. Below, we demonstrate a version of SEALS using a VAE as the feature extractor to compute the KNN graph only once. From a later experiment, we found that VAE-SEALS is slower than DGAL in query time and worse in its query-vs-accuracy trade-off.

**Test accuracy vs. number of selected labels.** In Figure 6, we report the classical active learning trade-off between accuracy and training set size. DGAL is observed to be competitive with several SOTA benchmarks, in particular, BADGE of Ash et al. (2019a) and CoreSet of Sener & Savarese (2017a). SEALS, on the other hand, is observed to be lagging in several data sets in an early stage of active learning because its selection criterion relies on the nearest neighbors of the existing training set which does not diversify the restricted set enough. In fact, its accuracy in the late stage does not even get close to existing benchmarks because refinement is not occurring either in the nearest neighbor-based query criterion.

**Average query time.** In Figure 7, the DGMG's average query time is an order of magnitude lower than that of all benchmarks. Note that the RS has an almost zero query time (at most $10^{-3}$ secs) that is barely visible in the plots.

**VAE and graph construction time.** We provide running time for the VAE and graph construction in Table 2 in the appendix. As seen, even with the pre-processing time our method is still faster than other methods, in particular, faster than the SEALS algorithm **without** its additional pre-processing time. We note that the pre-processing time is a one-time procedure, while the query is a repeated process, depending on each case. Hence, total query time can increase significantly higher than the pre-processing time. Hence the advantage of reducing it even with the price of pre-processing time.

### 5.3 DGMG vs. VAE-SEALS in accuracy and query time

We compare DGMG and DGLC with VAE-SEALS (see Algorithm 3). In VAE-SEALS, instead of extracting the features of unlabeled data from the task model and re-computing its KNN graph at each round, we use the same VAE latent space as used in DGMG. We note that in the SEALS paper Coleman et al. (2019), the authors didn't specify a specific representation space in the algorithm. This ablation study verifies that our query component based on diffusion is outperforming the KNN criterion of SEALS. We provide the results for this experiments in Figure 8 below and figures 10 and 11 in Appendix D.1. This study provides different

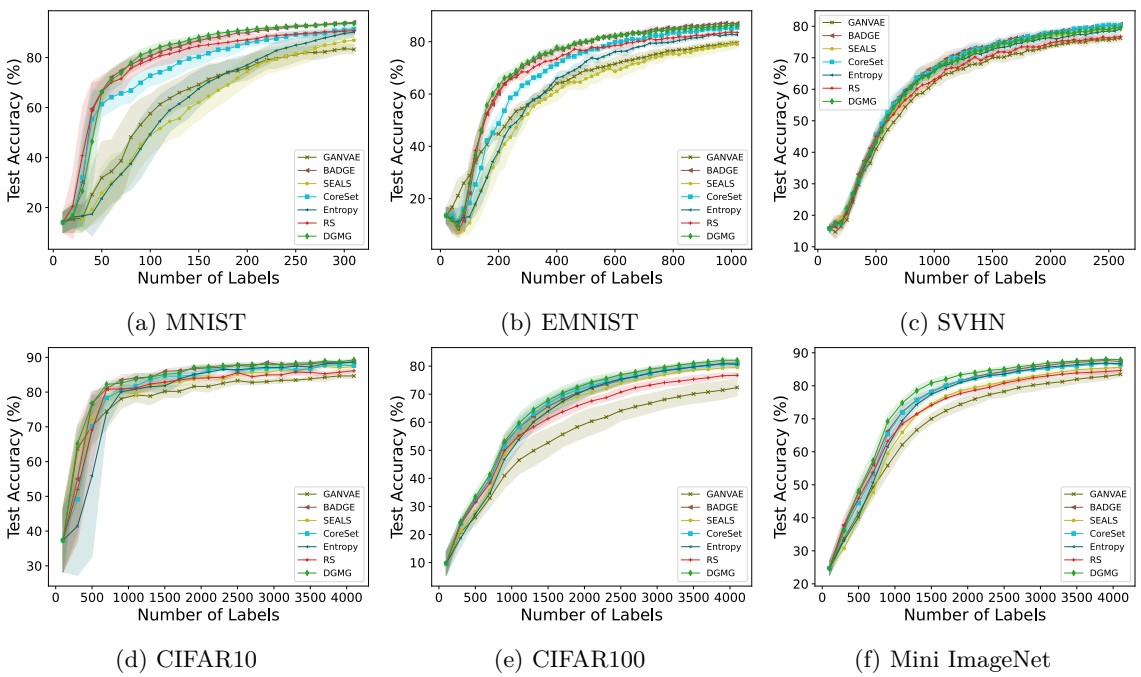

Figure 6: Plots of test accuracy vs. size of queried data. Each experiment is run at fixed query rounds for different methods and has been repeated 5 times.

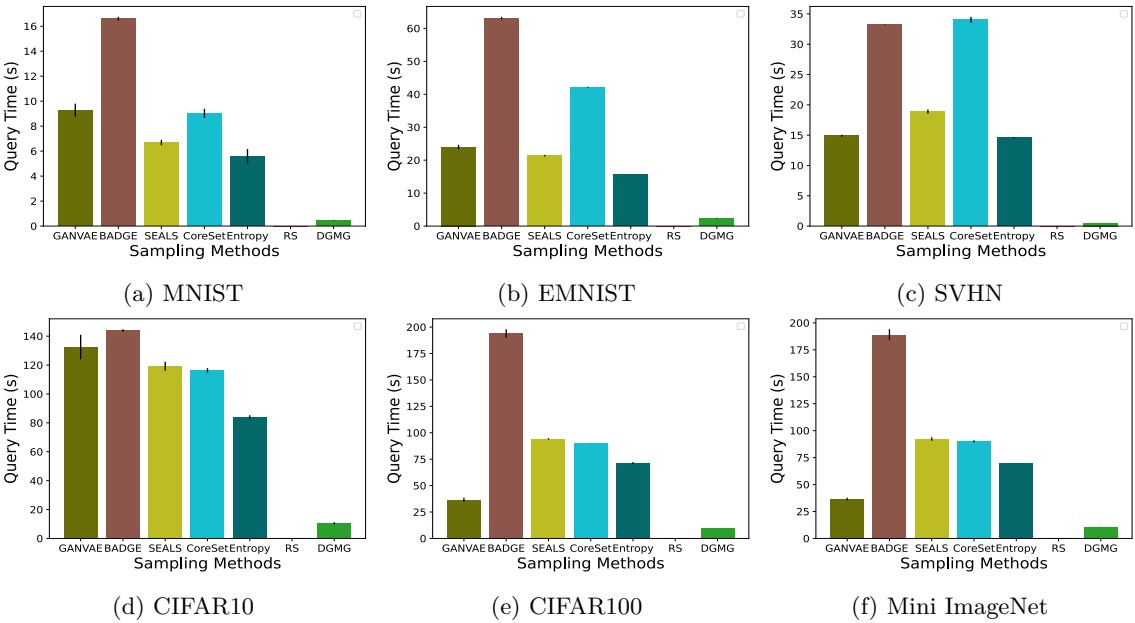

Figure 7: Query time for 6 benchmarks and DGMG, averaged over a fixed number of rounds for each method. The average time is averaged over 5 repetitions for each experiment.

experimentation on using a static pre-computed data representation with the KNN criterion, and its effect on the overall accuracy and query time of the methods. It also provides a view of the different versions of DGAL (DGMG and DGLC) and the advantage of the DGAL latent diffusion methodology over other restriction criteria. In particular, the comparison between VAE-SEALS and DGAL versions shows that DGAL is faster and achieves better accuracy. As seen in Figs. 8, DGMG has better accuracy than DGLC in most data sets.

In Figure 8, DGAL versions have an advantage over the VAE-SEALS. The VAE-SEALS can achieve similar accuracy with EMNIST, SVHN, CIFAR10, and Mini-ImageNet but uses more time than DGAL in these cases. It shows that the diffusion graph on latent space is more powerful than simply applying KNN on a latent space. On the other data sets SEALS doesn't achieve the accuracy that DGAL achieves within the same range of queries.

Comparing DGLC with VAE-SEALS in Figure 10, we observe the importance of pre-selection. The pre-selection of SEALS lacks diversity in the first few rounds of data acquisition. Under the same sub-sampling method (the least confidence sampling) after pre-selection, DGLC has an obvious advantage over VAE-SEALS at

---

**Algorithm 3** The VAE-SEALS strategy

**Input:** Labeled data $D_l$, unlabeled pool $D_u$, the initial VAE model $g$, the task model $f_\theta$, batch size $B$, round $R$, a query strategy $Q$, label $y$
**train** $g$ with $D_l$
**implement** the $k$-nearest neighbors structure $\mathcal{N}(\cdot, \cdot)$ on the latent space $g(D) = Z$
**initialize** the limited unlabeled pool as $D_u = \cup_{(Z,y) \in D_l} \mathcal{N}(Z, k)$
**for** $r = 1$ **to** $R$ **do**
    **active sub-sampling of a batch $B$ from $D_u$ with criterion Q and network $f_\theta$:** $D_Q = Q(f_\theta, D_u)$
    **annotate** $D_Q$
    **update** $D_l = D_l \bigcup D_Q$, $D_u = (D_u \setminus D_Q) \cup N(D_Q, k)$
    **train** $f_\theta$ with $D_l$
**end for**

---

the beginning of the active learning cycles, especially for MNIST, EMNIST, CIFAR10, and CIFAR100.

In Figure 11, we observe that our methods (DGMG, DGLC) win over the others in average query time. Due to the nature of the restriction process, all methods reduce the query time due to a smaller pool set. Nevertheless, as we see in Figures 8 and 10, while VAE-SEALS average query time may be similar to DGAL's, its accuracy trade-off with the query time is significantly worse than DGALs.

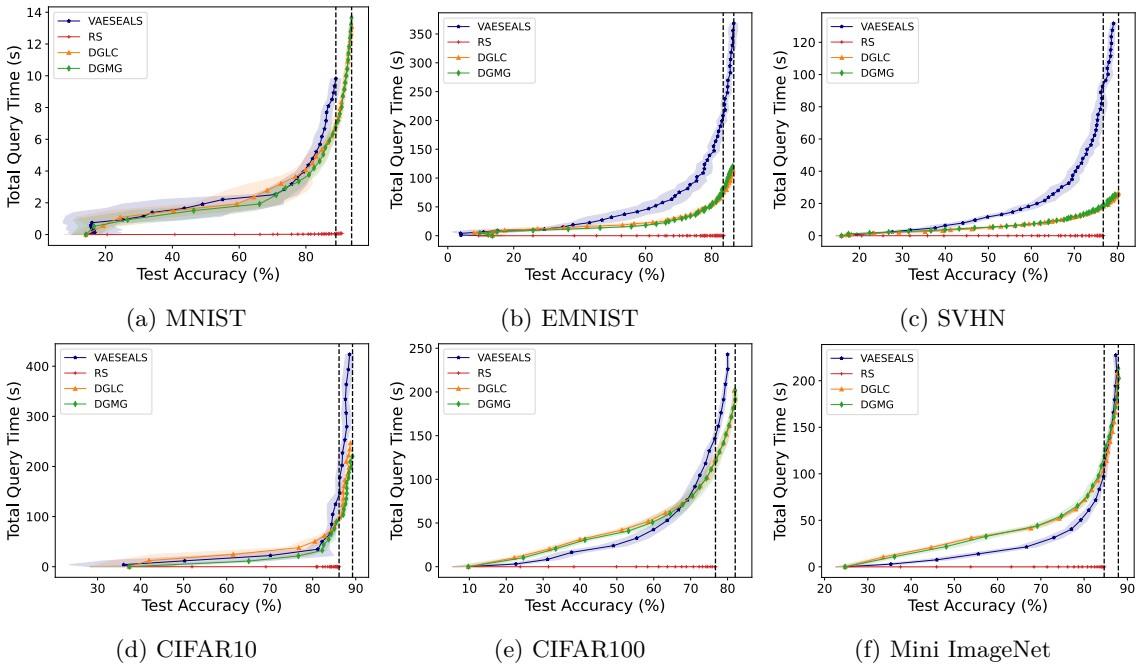

Figure 8: Test accuracy vs. total query time for VAE-SEALS, DGMG, DGLC, and Random (RS). The horizontal lines capture the highest and lowest test accuracy among all methods.

## 5.4 Ablation study

We conducted several ablation studies to investigate further on DGAL. First, we compare DGMG vs. non-DG-based confidence methods. The results show that the diffusion-based pre-selection contributes to a fast model improvement, especially in the early stage of querying. We also perform a study in which we substitute

various deep AL methods after our diffusion-based restriction of the candidate set in Figure 14. Methods combined with the VAE diffusion method are faster than running them alone. Due to space limitation, we provide plotted results in Appendix D.

### 5.5 Results on ImageNet data

The ImageNet Deng et al. (2009) is a well-known large-scale dataset in computer vision. It has approximately 1.3 million images with 1000 classes and an average size of $469 \times 387$ pixels per image. Due to the size of the data set, many AL algorithms did not finish execution in reasonable time. We therefore present in Figure 9 the results of selected AL methods as well as the DGMG. As observed, DGMG achieves better accuracy than VAE-SEALS and Random sampling. In these multi-class problems, the accuracy difference is significant. We run the experiment with a query size of 1000, and restriction to 20,000 in total query labels with $K = 10, T = 5$.

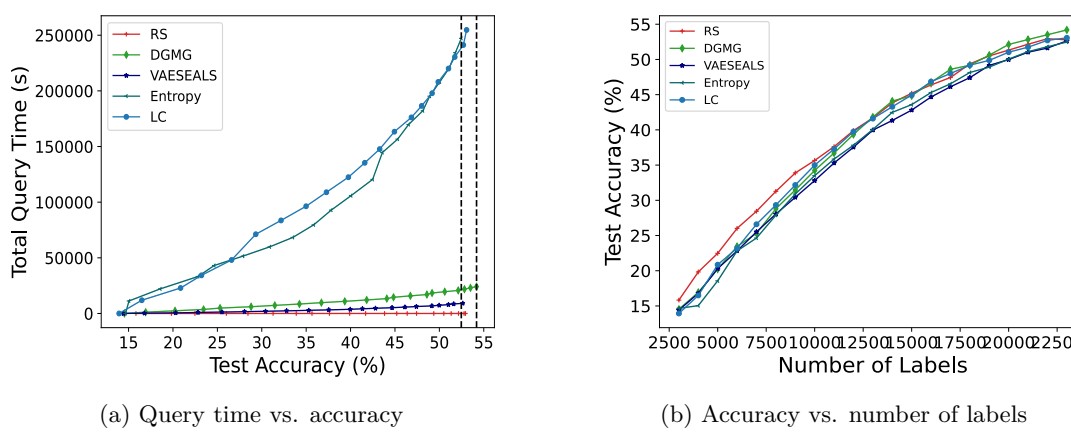

(a) Query time vs. accuracy  (b) Accuracy vs. number of labels

Figure 9: Experiments on the ImageNet data.

## 6 Conclusion

We proposed a diffusion-based sub-sampling method DGAL for efficient active learning. Unlike most active learning methods that feed the entire candidate pool set into the task model at each round, we first apply a diffusion model for pre-selection of the query set and then apply a deep active learning criterion on the subset for final label acquisition. Our method outperforms others in both test accuracy and query time in various experimental settings and diverse data sets. We show an order of magnitude acceleration in the query time compared to all other benchmarks. In the future, we plan to explore unsupervised representation learning schemes for improving latent space where label diffusion can be more efficiently applied.

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
