# A    Experimental settings

In the choice of network architectures, we employed different deep neural architectures: CNN, ResNet18, and small vision transformers (ViTs), as presented in Table 1. If not specifically mentioned otherwise, we optimized the network for the cross-entropy loss with SGD Kiefer & Wolfowitz (1952); Sutskever et al. (2013) optimizer at the learning rate 1e-3, momentum 0.9, weight decay 5e-4 for non-pre-trained models. The Adam Kingma & Ba (2014) optimizer with learning rate 1e-3 is used for pre-trained models. For methods that include VAE training, we use Adam optimizer with a 5e-4 learning rate by default. For all experiments, we adopted an early stopping criterion for epochs iterations with patience 10. For all the methods that need a data embedding feature space (i.e. CoreSet, SEALS, and BADGE) from the task model at each round, we added a linear layer at each network architecture of size 50 before the output layer, which is extracted by Pytorch Hooks. All the experiments were carried out on a NVIDIA RTX A6000. All the datasets and neural network information in the appendix follow the same details as shown in Table 1 in the main text.

Data and architecture details are provided in table Table 1.

Table 1: A summary of data and experimental settings that we used in our paper. Here 'Pre_n' is the size of a restricted subset $D'_{sub}$ in Algorithm 2. 'Budget' is the number of queried data points per round, i.e., the size of $D^\star$. 'Round' stands for the total sampling rounds.

| Dataset | Pool size | Label size | Input | Initial # of data | DimVAE | Pre_n/Budget /Round | Architectures | Pre-trained | Test size |
|---|---|---|---|---|---|---|---|---|---|
| MNIST | 60,000 | 10 | 28x28 | 10 | 50 | 100/10/30 | CNN | False | 10,000 |
| EMNIST letter | 124,800 | 26 | 28x28 | 20 | 256 | 200/20/50 | ResNet18 | True | 20,800 |
| SVHN | 73,257 | 10 | 32x32 | 100 | 100 | 500/50/50 | ResNet50 | False | 26,032 |
| CIFAR10 | 50,000 | 10 | 32x32 | 100 | 100 | 5k/200/20 | ResNet50 | True | 10,000 |
| CIFAR100 | 50,000 | 100 | 32x32 | 100 | 100 | 5k/200/20 | ViT-Small | True | 10,000 |
| Mini-ImageNet | 48,000 | 100 | 84x84 | 100 | 100 | 5k/200/20 | ViT-Small | True | 10,000 |
| ImageNet | 1,281,167 | 1000 | 224x224 | 100 | 100 | 10k/1000/20 | ViT-Small | True | 50,000 |

**Latent space generation.**    To get a representative latent space from data, we use a ResNet18-based encoder for CIFAR100 and MiniImageNet and a CNN-based VAE for the rest of the datasets. We use an Adam optimizer with a learning rate of 5e-4 and epoch 300. An early stopping criterion is used with patience 20. For the training of GANVAE, we use the same VAE and discriminator with learning rate 1e-4, total epochs 100, and no early stopping.

# B    Variants of multi-class extension

We propose the following other possible formulations to extend the diffusion-based active learning criterion to the multi-class setting.

**One-vs-all approach**    In the one-vs-all setting the batch is queried according to

$$\hat{X} = \arg\min_{i \in X_u}^{B} q\left(\chi_{:,i}^{(T)}\right)$$

for some chosen function $q$ which measures a notion of uncertainty at point $x_i$, given the matrix $\chi^{(T)}$. In Section 4.2, we chose $q$ to be

$$q\left(\chi_{:,i}^{(T)}\right) = \min_{c \in [C]}\left|\chi_{c,i}^{(T)}\right|, \; or \; q\left(\chi_{:,i}^{(T)}\right) = \left\|\chi_{:,i}^{(T)}\right\|_p$$

for some $p \in [1, \infty]$.

**Multivariate diffusion approach**    Moving from the one-vs-all approach, we can perform the query as follows, using the property that $M$ is a stochastic matrix. For each data point $x_i$, we propagate a probability

vector $\chi_i^{(t)} \in \Delta_C$. This vector can be initialized as

$$
\chi_{i,c}^{(0)} = \begin{cases} 1 & \text{if } i \in \mathcal{X}_\ell \text{ and } c = y_i \\ 0 & \text{if } i \in \mathcal{X}_\ell \text{ and } c \neq y_i \\ \frac{1}{C} & \text{otherwise} \end{cases}
$$

We can therefore diffuse the matrix aggregating the signal for all the points, $\chi^{(t)} \in \mathbb{R}^{N \times C}$, and diffuse it as in the binary case:

$$
\chi^{(t)} = M\chi^{(t-1)}, \qquad \chi^{(t)}|_{\mathcal{X}_\ell} = \chi^{(0)}|_{\mathcal{X}_\ell}
$$

Since $M$ is stochastic, it holds that $\chi^{(t)} \in \Delta_C$ at each iteration $t$. Therefore we can interpret each vector $\chi_i^{(t)}$ as a probability vector of the data point $x_i$ belonging to different classes, obtained by the diffusion above. It therefore makes sense to choose the points to query as

$$
\hat{X} = \arg\min_{k \in \mathcal{X}_u}^B q\left(\chi_i^{(T)}\right)
$$

where $q : \Delta_C \to \mathbb{R}$ is some measure of uncertainty. Possible choices include:

- Uncertainty: $q(p) = p_{c^*}$, where $c^* = \arg\max_c p_c$;

- Margin: $q(p) = p_{c^*} - p_{c_2^*}$, where $c^*$ is defined as above and $c_2^* = \arg\max_{c \in [C] \setminus \{c^*\}} p_c$;

- Negative entropy: $q(p) = \sum_{c \in [C]} p_c \log p_c$.

## C   VAE training and graph building time

Here we provide the total training time for VAE and graph construction time with latent variables.

Table 2: A summary of the VAE training time and graph building time for DGAL (measured in seconds).

| Dataset | VAE training | Graph construction |
|---|---|---|
| MNIST | 182 | 139 |
| EMNIST letter | 478 | 731 |
| SVHN | 276 | 207 |
| CIFAR10 | 208 | 108 |
| CIFAR100 | 1134 | 125 |
| MiniImageNet | 864 | 133 |

## D   Extended experiments and ablation study

### D.1   DGMG vs. VAE-SEALS

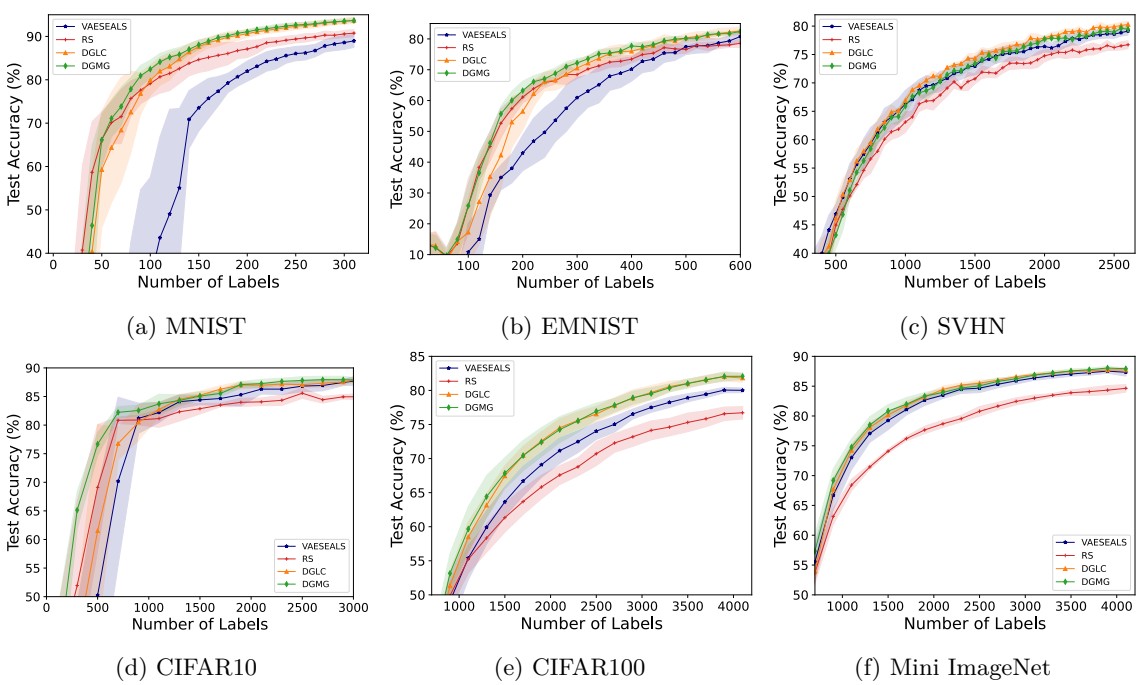

Figure 10: Test accuracy vs. size of queried data for VAE-SEALS, DGMG, DGLC, and Random Sampling (RS) over a limited total number of labels. Each experiment is run at fixed query rounds for different methods and repeated 5 times.

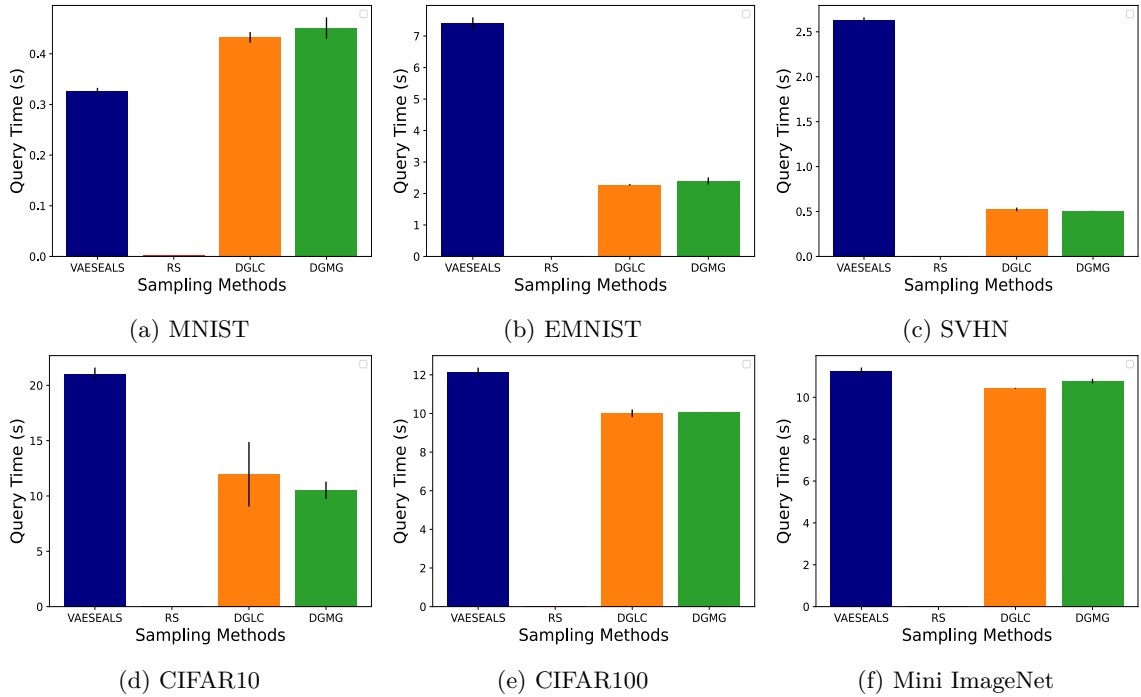

Figure 11: Query time for DGAL methods, VAE-SEALS, and Random Sampling (RS), averaged over a fixed number of rounds for each method. The average time is averaged over 5 repetitions for each experiment.

## D.2 DGMG vs. non-DG based confident methods

In this section, we compare DGMG with RSMG, Margin, and the baseline Random Sampling (RS). In RSMG, the pre-selection is random sampling while in DGMG it's based on label diffusion.

In Figure 12 we plot the test accuracy vs. size of queried data and observe DGMGs advantage at the early stage of learning. For example, in Figure 12d, DGMG is above RSMG and RS from 100 to 700 at the total number of labels. Similar trend exists also in Figure 12b, Figure 12c, and Figure 12e. It demonstrates how the latent diffusion graph pre-selects better than random sampling. In Mini-ImageNet, DGMG is close to RSMG, perhaps because the random baseline is a good explorer criterion. Margin sometimes achieves worse results in early learning. Yet it achieves similar accuracy as in DGMG at later stages of learning as it essentially uses a similar criterion but without restriction. Yet, Margin's query time is worse than DGMG and RSMG as seen consistently in Figure 13.

In Figure 13 we provide the average query time for each method. We observe an advantage for DGMG and RSMG in query time in comparison with Margin, mostly due to the efficiency of the restriction method. DGMG uses about 1/8th time of Margin's while achieving better test accuracy. While RSMG uses short query time, DGMG, in most cases, trades better additional time for better accuracy, as seen in Figure 12.

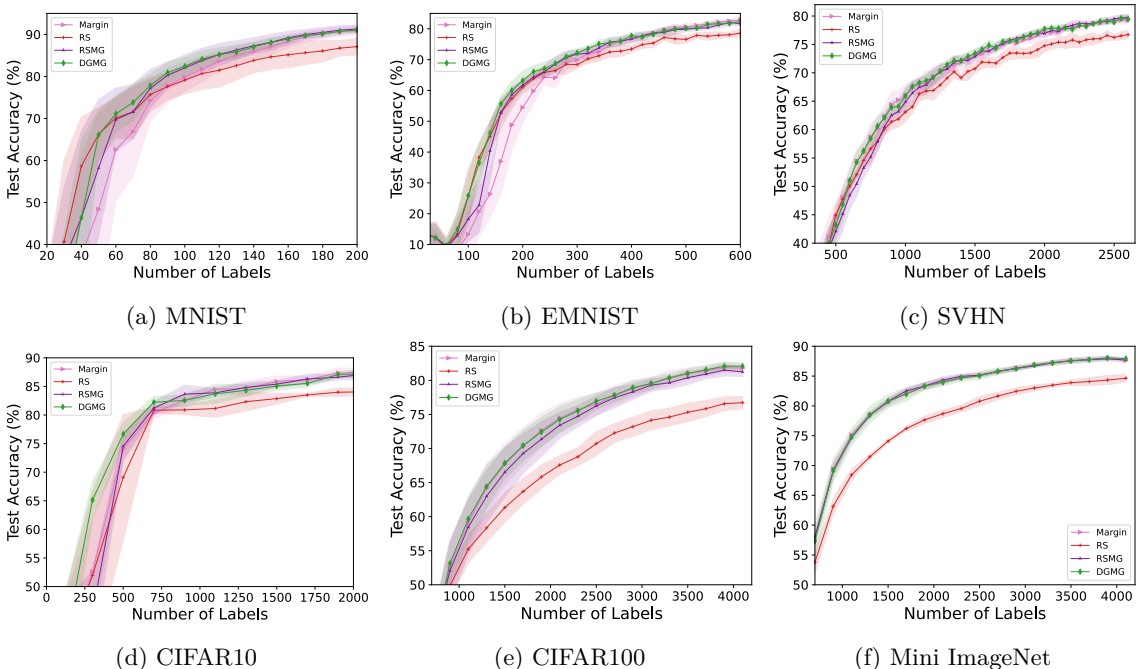

(a) MNIST      (b) EMNIST      (c) SVHN

(d) CIFAR10      (e) CIFAR100      (f) Mini ImageNet

Figure 12: Test accuracy vs. size of queried data for 3 baselines and DGMG. Each experiment is run at fixed query rounds for different methods and has been repeated 5 times.

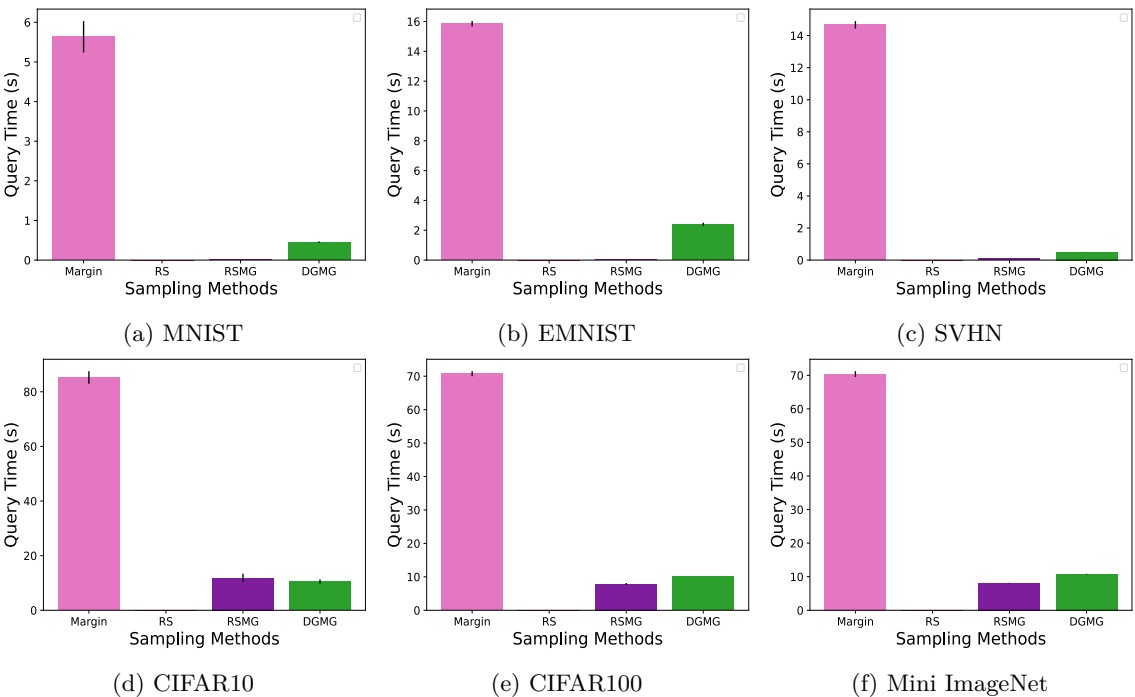

Figure 13: Query time for 3 baselines and DGMG, averaged over fixed rounds for each method. The average time is averaged over 5 repetitions for each experiment.

## D.3 DGAL with different deep AL criteria

We provide results of using additional AL criteria in the deep net architecture to test the acceleration based on the VAE and graph diffusion-based restriction.

It can be seen that DGBADGE may reach accuracy similar to DGMG but the trade-off with query time still renders it slower even on the restricted set in some of the tested data sets. For DGCoreSet, the situation is even worse as we can see a slower query time trade-off with accuracy consistently for all data sets.

As seen deep net AL criteria that are fast to compute and accurate enough are still showing the best tradeoff. this is probably due to the informative selection in the graph diffusion AL. Nevertheless, all methods combined with VAE diffusion active learning are faster than running the active learning method alone, as seen in Fig. 5

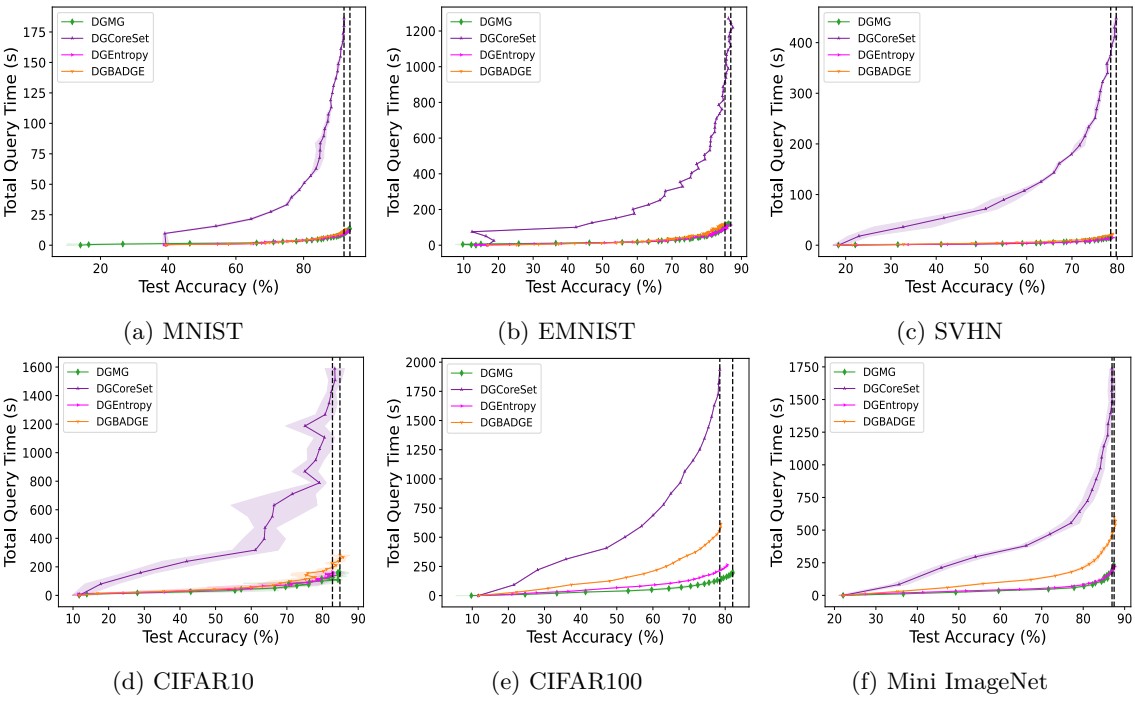

(a) MNIST  (b) EMNIST  (c) SVHN

(d) CIFAR10  (e) CIFAR100  (f) Mini ImageNet

Figure 14: Query time for 3 baselines and DGMG, averaged over fixed rounds for each method. The average time is averaged over 5 repetitions for each experiment.

## D.4 Study of nearest neighbor parameter in DGMG

We conducted a KNN ablation study to examine the effect of changing the number of neighbors in the diffusion graph. As observed, the accuracy of our method is not affected by the parameter $K$ for several sampled data sets that we examined in Figure 15. Changing $k$ will affect the query time, as the diffusion kernel will become denser which will result in longer multiplication times. However, yet in terms of active learning, the accuracy shows fairly robust behavior.

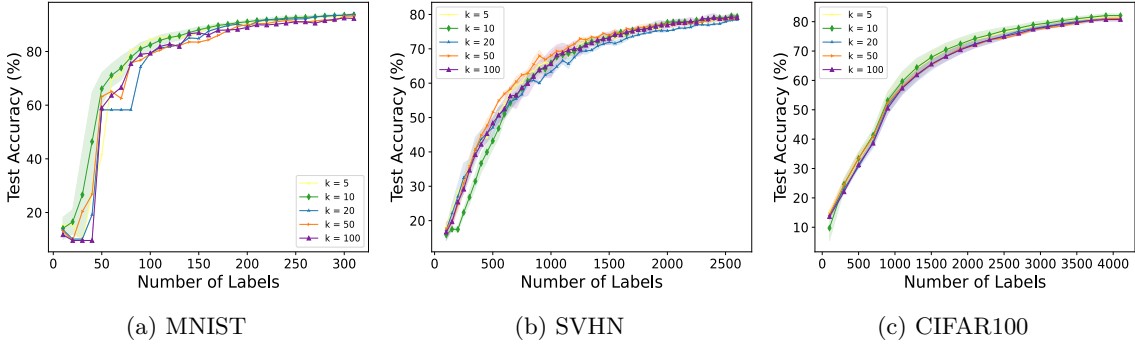

(a) MNIST  (b) SVHN  (c) CIFAR100

Figure 15: Query time for 3 baselines and DGMG, averaged over fixed rounds for each method. The average time is averaged over 5 repetitions for each experiment.