# OpenReview forum: "Accelerated Deep Active Learning with Graph-based Sub- Sampling"
_TMLR — Accepted by TMLR_

### Review · Reviewer_zj16 · 2024-02-20

**Summary Of Contributions:**

This paper proposes an active learning method based on a deep generative model VAE with a Graph Diffusion Method, which is one of the main contributions of this paper is the overall pipeline that is a stack of existing methods. The authors provide detailed ablation studies replacing components. The effectiveness of the proposed method was verified by a better trade offs between query-time and accuracy.

**Audience:**

Yes

**Broader Impact Concerns:**

None.

**Claims And Evidence:**

Yes

**Requested Changes:**

1. As mentioned in the weakness section, graph diffusion, and graph-based semi-supervised learning have been extensively studied in the literature. The authors need to compare the proposed method with existing methods regarding not only performance but also their pipelines to show the novelty of the proposed method in this paper. The role of generative modes, e.g., VAE, in this work is nothing but similarity computation, and it is not clear why the proposed method is computationally faster than other methods. All the graph-diffusion-based methods that use efficient matrix multiplication will have similar advantages since they have a short execution time per iteration. Figure 5 shows nothing but the proposed method. As shown in Figure 9 in the appendix, the proposed method does not show a relatively smaller improvement (not order of magnitude) with respect to the number of labels.

**Strengths And Weaknesses:**

Strengths
----

1. The query time has been improved compared to baselines. As the goal of this work, the proposed method shows superior performance. Figures 5 and 6 show that the proposed method shows improved query time, achieving better accuracy within a much shorter query time.
2. This paper is well-organized, and according to the author's changes, the readability of this manuscript has been improved, including notations, definitions, equations, and discussion about VAE.
3. Analysis for hyperparameters is provided in Section B. Graph diffusion methods are quite sensitive to hyperparameters.

Weaknesses
----

1. Although the overall pipeline has been shown effective, technical contributions of this paper is quite limited. No new component. The ablation studies mainly compared the components brought from previous works in the literature. For instance, the uncertainty was measured by existing and straightforward ways after graph diffusion.
2. Graph diffusion has been extensively studied in the literature. The authors used the existing concepts for semi-supervised learning. However, Graph Diffusion, or label smoothing on graphs are not a new concept. The authors should provide the related works graph diffusion for semi-supervised learning.

---

### Review · Reviewer_WaFV · 2024-02-24

**Summary Of Contributions:**

This work aims to improve the efficiency of deep active learning through utilizing the pre-trained VAE on the unlabeled data and an efficient query method on the hidden representations of the pre-trained VAE. This new method was compared to multiple earlier methods on several benchmarks. The authors show that their method (DGAL) outperforms existing methods on querying time while keeping a similarly high performance. They also provide ablation studies to support the design of their method.

**Audience:**

Yes

**Broader Impact Concerns:**

No clear negative society impacts from this.

**Claims And Evidence:**

Yes

**Requested Changes:**

See the weakness.

**Strengths And Weaknesses:**

**Strengths:**

The empirical evaluation results presented in this paper indicate that the new method yields great acceleration on the querying action. The authors have also evaluated their method on multiple benchmarks to show that this acceleration is significant across several scenarios. Different ways of active learning are also tested and the proposed method seems to outperform existing methods on querying time and achieves similar performance.

**Weaknesses:**

1.	The use of VAE may make this method unable to scale to large-scale and high-resolution datasets. More specifically, how important is the representation quality of the pre-trained VAE for the good performance of the algorithm? The authors mainly evaluated their methods on datasets with low resolution (the highest resolution tested is 84 * 84). VAE will have worse representation quality for larger datasets with higher resolution, such as the full ImageNet. What performance curve will this method yield on this type of datasets?


2.	How is the query time defined? I cannot find a definition for this important metric in the paper. Can the query time of other methods be reduced by applying methods like parallel computing?

3.	The discussion or conclusion section is missing. The authors miss the opportunity to discuss potential limitations of their methods.

4.	The writing needs to be improved. There are many typos in the paper, such as “bay be” instead of “may be” in the first paragraph of Sec. 4. Many sentences are also ungrammatical, such as missing “the” in some sentences.

---

> ### Author Response · Authors · 2024-04-09
> **Response to reviewer WaFV**
>
> We thank the reviewer for his\hers important remarks and suggestions to improve our manuscript. We also thank him\her for the strong support of our reported results, and the appreciation of the variety of experiments we conducted. We addressed all of his points below.
>
> 1.**The use of VAE may make this method unable to scale to large-scale and high-resolution datasets. More specifically, how important is the representation quality of the pre-trained VAE for the good performance of the algorithm? The authors mainly evaluated their methods on datasets with low resolution (the highest resolution tested is 84 * 84). VAE will have worse representation quality for larger datasets with higher resolution, such as the full ImageNet. What performance curve will this method yield on this type of datasets?**
>
> The representation quality of the VAE is important but not critical. Most importantly, the VAE is an unsupervised representation which makes it a method of choice because other representations may involve supervision. In this paper we are proposing an active learning framework, which by its definition aims to reduce the amount of supervision. Therefore, our choice of VAE is justified for the sake of reducing the data annotation burden.
>
> Indeed, if other adequate pre-trained representation, which do not require additional supervision can be used, they may yield useful representations. However, in this paper we do not explore representation selection, but propose an overall architecture that allows to accelerate active learning query time. We have incorporated that in our revision, and we further amplify this point in the paper.
>
> In addition, and in response to your request and reviewer’s AgA4 remark, we added in appendix E.5 results of running DGMG on 1.3M ImageNet (469x387).  It shows that DGMG together with VAE-SEALS are scalable, yet, DGMG achieves the highest accuracy among all methods reported, thanks to its graph based active learning. Moreover, Least Confidence and Entropy, using standard AL criteria (not graph based), with no pool set restriction, are showing much worse tradeoff. Other methods, (BADGE, CorseSets, GANVAE) that use computationally heavy criteria, without any graph-based restriction of the pool, didn’t even finish execution until now.
>
> 2.**How is the query time defined? I cannot find a definition for this important metric in the paper. Can the query time of other methods be reduced by applying methods like parallel computing?**
>
> We added this definition to our paper in section 3. Parallel computing may be applicable for certain simple query criteria (like Entropy, or Least Confidence) but not for all the tested methods. For some algorithms there are dependencies that cannot be parallelized, for example in BADGE, or CoreSet algorithms. However, the main problem with simple query criteria is that their accuracy is limited as they do not tradeoff exploration and refinement, as our diffusion based method.
>
> 3.**The discussion or conclusion section is missing. The authors miss the opportunity to discuss potential limitations of their methods.**
>
> Thank you. This section was missing mostly due to space reasons. We have now added the conclusion section.
>
> 4.**The writing needs to be improved. There are many typos in the paper, such as “bay be” instead of “may be” in the first paragraph of Sec. 4. Many sentences are also ungrammatical, such as missing “the” in some sentences.**
>
> Thank you, we went through the paper back-to back and corrected these issues.

---

### Review · Reviewer_AGA4 · 2024-03-19

**Summary Of Contributions:**

This paper proposes a new way to efficiently sample data points for active learning. Unlike past methods that need to scan over the entire dataset (hence intractable), the proposed method use a pre-trained VAE to restrict the size of the pool to a much smaller candidate set. This method demonstrates a better empirical efficiency.

The key contribution of this paper lies in the incorporation of the pre-trained VAE to embed raw data to a latent space, where a method caeed latent graph diffusion can be applied. The graph is computed by a similarity metric among the nearest neightborhood. Then a graph diffusion process is applied on this graph to propagate the label information.

Overall, I think this paper introduces a simple graph-based sampling method to address the inefficiency of the label acquisition problem in active learning. Despite being a bit unnecessarily complicated, this method is able to demonstrate better efficiency over baselines.

**Audience:**

Yes

**Claims And Evidence:**

No

**Requested Changes:**

Generally, I think the novelty of this method is incremental. But the novelty is not the primary focus of TMLR, so I will just focus on its claims.

- The first claim that this method is scalable and efficient does not seem to hold with its current experiments (only CIFAR, MNIST and Mini-imagenet are used).

- The speed-up is a magnitude better seems to hold. But Figure 5 should be modified to better reflect this claim.

For the other concerns, see the "Strengths And Weaknesses" section.

**Strengths And Weaknesses:**

I think the idea is pretty straightforward. Essentially, a pre-trained VAE is used for dimensionality reduction and then a sampling strategy that uses a graph diffusion process to propagate label information is applied.

- The method seems to be empirically effective, although I still found the proposed method unnecessarily complicated. Whether the graph diffusion really necessary in unclear to me.

- The method claims to address the intractability of large pool size, but the experiments also include datasets such as MNIST, CIFAR, Mini-imagenet. It is not very convincing to me that this method can really scale to millions of samples.

- As a minor point, the improvement looks a bit marginal to me. For Figure 5, it could be better to use logarithm X-axis to show the improvement is really a magnitude of difference.

- The ablation study does not well justify many of the design choices. Besides, hyperparameters should also be ablated. For example, different size of the neightborhood set should be tested.

---

> ### Author Response · Authors · 2024-04-09
> **Authors response to review AGA4**
>
> We thank the reviewer for his\hers important remarks and for his supporting comments on our empirical results. Below we address his\hers comments and the changes we introduced to our manuscript following his suggestions.
>
> **The method seems to be empirically effective, although I still found the proposed method unnecessarily complicated. Whether the graph diffusion really necessary in unclear to me**
>
> We have shown in fig 5 and in our comparison with SEALS that the use of the graph diffusion provides a far better selection of queries and therefore higher accuracy. SEALs is using a k-nearest neighbors to select the restricted pool set. As seen in Fig 5 and Fig 8 its trade-off of query time vs accuracy is inferior to DGMG. Any other method that uses a different query criteria than diffusion is also falling behind on accuracy and its trade-off with query time.
>
> The diffusion criterion is critical for the success of the method, and not an unnecessary complication. Its key advantage is in the exploration-refinement sampling that uniquely characterizes it, and allows it to outperform other criteria. Other methods, except BADGE, do not trade exploration with refinement.
>
> **The method claims to address the intractability of large pool size, but the experiments also include datasets such as MNIST, CIFAR, Mini-imagenet. It is not very convincing to me that this method can really scale to millions of samples.**
>
> We have now added to appendix E.5 the full ImageNet data set of 1.3M data points. In the two weeks that we had for this rebuttal we managed to run DGMG, the random sampling baseline, VAESEALS  [Coleman et al] that also uses a precomputed representation, and entropy. Other methods, such as BADGE and COARESET, did not finish running on ImageNet with our compute resources in the two weeks’ timeline. This also proves the importance of our method as an enabler of active learning in large scale problems. The results show that our methods query time is affected by the size of the pool set, but it achieves the highest accuracy in the experiment we performed and is faster than 4 other AL methods we tested.
>
> **The ablation study does not well justify many of the design choices. Besides, hyperparameters should also be ablated. For example, different size of the neighborhood set should be tested**
>
> We added the neighborhood size to the ablation study. We are not sure what other parts of the ablation study the reviewer sees as missing. But we would like to better explain what the existing ablations studies already provided show:
> 1. Figs 9-10 we show that different AL criteria for restriction such as the one in VAESEALS, even though are using a precomputed representation, do not show better results than our selection of the diffusion-based restriction method.
> 2.	Figs 11-12 shows that using the same query method that we use (‘margin’) without the restriction step does not yield better results.
> 3.	Fig 13 shows that the selection of the margin query method on top of the diffusion-based restriction makes margin a better choice than other classical AL criteria.
> 4.	Finally we added the nearest neighbor ablation study, as per the reviewers request to show that the size of the neighbors set is robust with respect to the method accuracy, which clearly it can introduce higher query time as k grows. By that we justified our selection of k=10 in our experimental design
>
> As for other parameters, such as T, we provided an analysis and theoretically-based selection for these parameter values in appendix B.
>
> **The first claim that this method is scalable and efficient does not seem to hold with its current experiments (only CIFAR..).**
>
> We added 1.3M ImageNet experiment to the appendix E.5 and figure 15. It shows that DGMG together with VAE-SEALS are scalable. Yet, DGMG achieves the highest accuracy among all methods reported for the query time reported, thanks to its graph based active learning. Moreover, Least Confidence and Entropy, using standard AL criteria (not graph based) are showing worse tradeoff. Other methods, (BADGE, CorseSets, GANVAE) that use computationally heavy criteria, without any graph-based restriction of the pool, didn’t even finish execution until now.
> To this end, for Imagenet we observe 3 scalability groups: restriction based active learning (VAESEALS, DGMG) at ~10^4 seconds. Standard, simple active learning (Least confidence and Entropy), with no pool-set restriction at ~2.5 x 10^5 seconds, and more complex active learning algorithms (BADGE, CoreSet, VAEGAN) that to this end exceeded 10^6 seconds, and haven’t finished execution yet.
>
> **The speed-up is a magnitude better seems to hold. But Figure 5 should be modified to better reflect this claim.**
>
> Indeed it holds. We introduced a modification to Fig 5 and in other images that present the tradeoff between query time and accuracy. We essentially flipped the axes and it is clear now that there is an order of magnitude improvement.

---

### Author Response · Authors · 2024-04-10
**Rebuttal summary**

We sincerely thank the reviewers for their important remarks and suggestions to improve our manuscript.
We have addressed all the points raised in the uploaded revised manuscript and in each of our responses.
We emphasis below the main points addressed in the rebuttal:

1.	Based on reviewer’s feedback, large scale ImageNet data set added (appendix E.5) to demonstrate the scalability of our method and SotA.
2.	In our responses, we emphasized again the importance of the graph diffusion-based pool set restriction over any other criteria. We also redraw the link between our arguments and the results of our experiments and the ablation study.
3.	We edited our results figure (e.g. Fig 5) to better reflect the order of magnitude improvement that our method exhibits.
4.	We provided an additional ablation study on the nearest neighbor hyper-parameter.

Please feel free to contact us regarding any further questions or clarification you may have.

Thank you,

The authors.

---

### Decision · Action_Editor_j6kQ · 2024-07-17

**Recommendation:** Accept as is

**Comment:**

Some concerns were initially raised about the empirical support for the claims regarding improved scalability on large scale datasets. During the review process, the authors added experiments on ImageNet that demonstrate the evidence still holds in the large scale setting. The reviewers were generally satisfied by this additional evidence.

The authors are encouraged to move relevant details about the time complexity (Appendix B) and the ImageNet results (Appendix E.5) to the main paper as these are important pieces of evidence supporting the claims of the paper. Additionally, there are some remaining (minor) formatting and grammar issues that should be corrected.

**Audience:**

Methods for improving the scalability of active learning are of general interest to the machine learning community.

**Claims And Evidence:**

This paper introduces a method called DGAL that uses VAE pre-training and latent graph diffusion to improve the scalability and efficiency of active learning. These improvements are demonstrated empirically on image classification datasets ranging from MNIST to ImageNet (see e.g. Figure 5 comparing the query time vs. test accuracy tradeoff against several baselines).

---

> ### Author Response · Authors · 2024-08-06
>
> Thank you for handling our paper. We've moved Appendix B and E.5 to the main text as suggested. We've also carefully revisited our paper and fixed all the typos. Please kindly let us know if further action is needed.